# Diagnosis of Visceral Leishmaniasis in an Elimination Setting: A Validation Study of the Diagnostic Algorithm in India

**DOI:** 10.3390/diagnostics12030670

**Published:** 2022-03-09

**Authors:** Kristien Cloots, Om Prakash Singh, Abhishek Kumar Singh, Anurag Kumar Kushwaha, Paritosh Malaviya, Sangeeta Kansal, Epco Hasker, Shyam Sundar

**Affiliations:** 1Unit of Mycobacterial Diseases and Neglected Tropical Diseases, Department of Public Health, Institute of Tropical Medicine, 2000 Antwerp, Belgium; ehasker@itg.be; 2Department of Biochemistry, Institute of Science, Banaras Hindu University, Varanasi 221005, India; opbhu07@gmail.com; 3Infectious Diseases Research Laboratory, Department of Medicine, Institute of Medical Sciences, Banaras Hindu University, Varanasi 221005, India; abhishek.nbu@gmail.com (A.K.S.); anuragmicro27@gmail.com (A.K.K.); paritosh_malaviya@yahoo.com (P.M.); sangeetakansalbhu@gmail.com (S.K.); drshyamsundar53@gmail.com (S.S.)

**Keywords:** visceral leishmaniasis, kala-azar, diagnostic algorithm, rK39 RDT, low incidence setting, elimination

## Abstract

Visceral leishmaniasis (VL) is on the verge of elimination on the Indian subcontinent. Nonetheless, the currently low VL-incidence setting brings along new challenges, one of which is the validity of the diagnostic algorithm, based on a combination of suggestive clinical symptoms in combination with a positive rK39 Rapid Diagnostic Test (RDT). With this study, we aimed to assess the positive predictive value of the diagnostic algorithm in the current low-endemic setting in India by re-assessing newly diagnosed VL patients with a qPCR analysis on venous blood as the reference test. In addition, we evaluated the specificity of the rK39 RDT by testing non-VL cases with the rK39 RDT. Participants were recruited in Bihar and Uttar Pradesh, India. VL patients diagnosed based on the diagnostic algorithm were recruited through six primary health care centers (PHCs); non-VL cases were identified through a door-to-door survey in currently endemic, previously endemic, and non-endemic clusters, and tested with rK39 RDT, as well as—if positive—with qPCR on peripheral blood. We found that 95% (70/74; 95% CI 87–99%) of incident VL cases diagnosed at the PHC level using the current diagnostic algorithm were confirmed by qPCR. Among 15,422 non-VL cases, 39 were rK39 RDT positive, reflecting a specificity of the test of 99.7% (95% CI 99.7–99.8%). The current diagnostic algorithm combining suggestive clinical features with a positive rK39 RDT still seems valid in the current low-endemic setting in India.

## 1. Introduction

Visceral leishmaniasis (VL), also called kala-azar on the Indian subcontinent, is a parasitic, vector-borne, Neglected Tropical Disease that is fatal if not treated in a timely fashion. Since the kala-azar elimination initiative was launched on the Indian subcontinent in 2005, an important reduction in the number of VL cases has been observed in the region. In India, reported cases of VL dropped from 32,803 in 2005 to 2033 in 2020, translating into an annual incidence below the elimination threshold of 1 per 10,000 population in 98% (617/633) of blocks in 2020 [1]. This low-incidence setting brings along new challenges, one if which is the validity of the current diagnostic algorithm [2,3].

Visceral leishmaniasis usually presents with a spectrum of rather non-specific clinical symptoms. Therefore, the National Guidelines in India state that a diagnosis of VL should be based on a combination of suggestive clinical signs (mainly fever for more than two weeks and hepatosplenomegaly) and a positive serological and/or parasitological test [4]. As parasitological tests require well-equipped medical and laboratory facilities, serology is the test of choice in most settings, with the rK39 Rapid Diagnostic Test (RDT) being the most commonly used test by far. The rK39 RDT has an estimated sensitivity of 97% and a specificity of 90% on the Indian subcontinent [5]. However, the reliability of the test result, also called the validity of the test, also depends on the pre-test probability of the VL suspect to actually have VL, and, therefore, on the underlying prevalence of the disease in the community. The rarer the disease, the lower the probability that an individual testing positive with the test really has the disease (positive predictive value = PPV), and the higher the probability of a false positive result. On the other hand, the opposite is true for the negative predictive value, which tends to improve with declining disease prevalence. So far, the diagnostic algorithm has only been validated for high-incidence settings [6]. A 2014 systematic review estimated the positive predictive value of the rK39 RDT in clinical suspects to be 94% in a context with a pre-test probability of 60%, being reduced to 87% in a setting with a pre-test probability of 40% [5]. In the current low-prevalence setting, however, the pre-test probability—and therefore the PPV—can be expected to be even lower.

With this study, we aimed to assess the validity of the diagnostic algorithm for the diagnosis of VL in the current, low-endemic, peri-elimination setting in India, by assessing the positive predictive value of the diagnostic algorithm (suggestive clinical symptoms in combination with a positive rK39 RDT) by using a qPCR analysis on venous blood as the reference test. In addition, we estimated the specificity of the rK39 RDT.

## 2. Materials and Methods

### 2.1. Field Work and Study Sites

The study was conducted in the two VL endemic states of Bihar and Uttar Pradesh in Northern India, reporting an overall VL incidence rate in 2018 (at the start of the study) of 0.3/10,000 population (3423/118 million population) and 0.005/10,000 population (110/222 million), respectively [7,8]. However, VL is unevenly spread and incidences at a district or sub-district level can be much higher. Participants were recruited through two parallel approaches.

Inclusion of VL patients. A total of six primary health care centers (PHCs) in Bihar and Uttar Pradesh were selected for VL patient recruitment. A PHC covers the primary health care needs of a block, which is the sub-district-level administrative unit with a population of approximately 100,000–200,000 people. The selection of PHCs was done based on the VL case load in the respective block in the five years prior to the study, aiming for a combination of PHCs with a high (annual VL incidence >1/10,000 population) and a low (annual VL incidence <1/10,000 population) VL burden. PHCs selected for patient recruitment were those of the high-incidence blocks of Garkha (Saran district, Bihar), Dariapur (Saran district, Bihar), and Sonepur (Saran district, Bihar), and of the low-incidence blocks of Kanti (Muzaffarpur district, Bihar), Bairia (Ballia district, Uttar Pradesh), and Bansdih (Ballia district, Uttar Pradesh). Detailed information on the annual VL incidence in each of the selected blocks in the five years prior to the study can be found in Appendix A. The geographical location of the selected PHCs is illustrated in Figure 1. All incident VL patients diagnosed between 10 June 2019 and 16 April 2021 in the selected PHCs, using the National Guideline recommended standard diagnostic approach of suggestive clinical symptoms (mainly fever for more than two weeks and hepatosplenomegaly) in combination with a positive rK39 RDT (on either a capillary or venous blood), were requested to provide a 2 mL venous blood sample to be tested via quantitative polymerase chain reaction (qPCR) as a proxy for *Leishmania donovani* parasite load in the blood. All patients were also tested for HIV infection as part of the routine VL case management as recommended in the National Guidelines [4]. Information on demographic characteristics and VL-related topics was collected through an Android application on a smartphone by the PHC staff, sending the data to a secured server on a daily basis. Once the venous blood sample was collected, it was labelled with a unique barcode that linked the sample to the correct participant and was then stored at 2–8 °C until transportation to the laboratory facilities of Banaras Hindu University (BHU), Varanasi, India, where it was stored at −20 °C until qPCR analysis.

Inclusion of non-VL cases. In addition, a serological survey was carried out in seven clusters (part of a village) in Bihar and Uttar Pradesh between 13 May 2019 and 29 December 2020. These clusters were selected based on the VL caseload in the 15 years prior to the field work, and included currently endemic (ongoing VL cases in each of the last three years), previously endemic (no cases in the last 4 years, but cases reported in the years before), and non-endemic clusters (no VL cases reported in the last 15 years). The cluster-specific VL case load between 2015–2018 and population data can be found in Appendix A. After obtaining informed consent (assent for minors ≥12 years, in combination with consent from the guardian), all inhabitants above the age of 2 years were requested to provide a finger prick blood sample to conduct an rK39 RDT test on the spot. Information on demographic parameters, as well as VL history, was collected using an Android application on mobile phones, and sent to a secured server at the end of the day. Individuals with a positive rK39 RDT were asked to provide a 2ml venous blood sample for testing with qPCR. Once the venous blood sample was collected, it was labelled with a unique barcode that linked the sample to the correct participant and was then stored at 2–8 °C until transportation to the laboratory facilities of BHU, where it was stored at −20 °C until qPCR analysis. Participants with a history of fever for at least two weeks in combination with a positive rK39 RDT were categorized as VL case and were therefore excluded from further analysis.

### 2.2. Sample Testing and Laboratory Analysis

rK39 RDT tests (InBios, Seattle, WA, USA) were conducted according to the manufacturer’s instruction, with the exception that capillary blood was used instead of serum, corresponding to how the test is being used in most clinical settings in reality. Concordance between test results on whole blood and serum samples is assumed to be excellent [9].

Confirmation of the diagnosis of VL was done using qPCR on venous blood samples. The qPCR procedure was carried out at the laboratory facilities of BHU as described previously [10]. Briefly, DNA was isolated from 200 µL of blood using a QIAmp DNA blood kit (QIAGEN, GmBH) according to the manufacturer’s manual. A TaqMan-based absolute quantification method was performed on each DNA sample to quantify the *Leishmania donovani* parasite DNA using *Leishmania*-specific kDNA primers on a 7500 Real time PCR System (ABI, Thermo Fischer Scientific, Massachusetts, MA, USA). The generation of amplification plots, standard curves, and dissociation stage analysis was done by ABI SDS software. The fluorescence intensity of each sample, which is proportional to the amount of DNA present, was expressed in terms of the PCR threshold cycle (Ct), defined as the number of PCR cycles required for the fluorescence. The number of *Leishmania* parasites was calculated and extrapolated by comparing the cyclic threshold (Ct) value of the sample with a standard curve (in a range of 10,000 to 0.01 parasite). The result was considered positive if the Ct value was ≤36.

### 2.3. Data Analysis

Data analysis was performed using R studio (version 2021.09.0). The positive predictive value (PPV) of the diagnostic algorithm was calculated as the proportion of qPCR positive individuals among incident VL cases diagnosed at the PHC level according to the diagnostic algorithm of suggestive clinical symptoms in combination with a positive rK39 RDT. The specificity of the rK39 RDT was calculated as the proportion of non-VL cases living in currently endemic, previously endemic, or non-endemic areas who were negative by rK39 RDT. To calculate the 95% confidence intervals around proportions, the exact method of Clopper and Pearson (binom.test function) was used.

## 3. Results

### 3.1. Findings among Newly Diagnosed VL Patients at the PHC Level

A total of 78 VL patients were diagnosed at the PHCs during the study period, of which 74 (95%) agreed to take part in the study. Most of the included VL patients were diagnosed in the Dariapur block (51/74). The median age of the included VL patients was 29 years (IQR 16–41y), and the majority were male (44/72 = 61%; *p* = 0.08). For two patients, no sex or age was recorded. None of the patients were co-infected with HIV. Out of the 74 VL patients, 70 tested positive with qPCR, yielding a PPV of 95% (70/74 = 95%; 95% CI 87–99%) of the diagnostic algorithm. The median parasitemia of the 70 qPCR-positive VL patients was 6844 parasite genome equivalents per ml of blood (PGE/mL) (IQR 988–44,464 PGE/mL), ranging between 0.3 and 1,814,246 PGE/mL. Participant characteristics and findings per PHC are summarized in Table 1. A boxplot illustrating the parasitemia distribution is provided in Appendix A.

### 3.2. Findings among Non-VL Cases

A total of 15,422 participants agreed to be included in the serological survey; 5528 were from currently endemic clusters, 6040 were from previously endemic clusters, and 3854 were from non-endemic clusters. The median age of the participants was 23 years (IQR 12–40y), and 47% (7218/15,422; *p* < 0.001) were male. None of the participants reported clinical symptoms suggestive for VL at the time of the survey; therefore, all participants were considered non-VL cases in the further analysis. Out of the 15,422 (0.25%) participants, 39 tested positive with the rK39 RDT, reflecting a specificity of 99.7% (95% CI 99.7–99.8%) for the rK39 RDT. Thirty-six (92%) of these rK39 RDT-positive participants agreed to be tested with qPCR, with 22 of them reporting a previous VL history and 14 reporting no such history. Overall, one out of 36 qPCR results was positive (2.78%; 95% CI 0.07–14.53%). The qPCR-positive participant resided in a previously endemic cluster, but did not report a history of VL. Cluster-wise characteristics and findings among the non-VL cases are summarized in Table 2.

## 4. Discussion

In this study, we found the PPV of the diagnostic algorithm (suggestive clinical symptoms in combination with a positive rK39 RDT) to be 95% (95% CI 87–99%) when using a qPCR analysis on peripheral blood as a reference test. In addition, we found the rK39 RDT to have a specificity of 99.7% (95% CI 99.7–99.8%). The diagnostic approach using clinical symptoms in combination with a positive rK39 RDT still seems valid in the current low-incidence setting in India.

The PPV of 95% observed in this study means that 5% of patients diagnosed with VL at the PHC level using the diagnostic algorithm could not be confirmed with qPCR. This implies that either qPCR failed to detect the existing *Leishmania* parasites in these patients, or the diagnostic algorithm falsely classified them as VL patients.

It is indeed possible that qPCR failed to identify 5% of VL patients. Although the sensitivity of molecular methods on peripheral blood is assumed to be excellent, 5% of false negatives would be in line with the sensitivity estimate of 93% reported in a 2014 systematic review [11]. Nonetheless, a more recent study conducted in Bangladesh, as well as two recent studies in India using the same qPCR protocol as used in the current study (conducted by the same laboratory), reported a 100% sensitivity of qPCR for the diagnosis of VL [12,13,14].

Alternatively, the diagnostic algorithm falsely classified 5% of clinical suspects as VL. Recent estimations of the PPV of the diagnostic algorithm in the literature are scarce. A 2014 Cochrane review estimated that with an rK39 RDT sensitivity and specificity of 97% and 90%, respectively, the PPV of an rK39 RDT among clinical suspects on the Indian subcontinent would be 94% only if the pre-test probability was 60% [5]. Although a pre-test probability of 60% seems high in the current low-endemic setting, it is possible that this is still a valid estimation at present if the rK39 RDT is only used at the end of a series of tests ruling out the more common causes of febrile splenomegaly in this region. To our best knowledge, however, recent estimations on rK39 RDT pre-test probabilities are not available. The PPV could remain similar at pre-test probabilities below 60% only if the specificity of the rK39 RDT, in reality, is higher than the 90% assumed in the 2014 review [5]. In this study, we found 39 out of 15,422 non-VL cases to be positive with rK39 RDT, yielding a specificity of at least 99.7% (95% CI 99.7–99.8%). Whether the positive rK39 RDT in these non-VL cases reflects true infection, or rather false positive results, is difficult to assess in the absence of additional markers of infection. Since one of these rK39 RDT positive non-VL cases was also positive with qPCR, however, it is likely that at least for this participant, the rK39 RDT reflected a true infection if not early—still asymptomatic—disease, which would increase the specificity of the rK39 RDT even further. Our findings are in line with those of a Phase III diagnostic accuracy study performed in India, reporting a specificity of the rK39 RDT of 97% [6]. The high specificity of the rK39 RDT might also explain why, despite the low prevalence of VL at present, the PPV of the diagnostic algorithm is still excellent.

A limitation of this study was that we used molecular testing (qPCR) instead of parasitology on splenic aspirates—which is generally accepted as the gold standard—for the confirmation of VL disease. The main reason for this were ethical concerns with regard to subjecting patients to invasive sampling, associated with life-threatening hemorrhage in an estimated 0.1% of interventions [15], for the sole aim of this study. While sensitivity is excellent and even suggested to be superior to that of parasitology [16,17], specificity of molecular testing for the diagnosis of VL remains a topic of debate. A 2014 review illustrated that the overall specificity of molecular methods to detect *Leishmania* parasites on peripheral blood was excellent (95.6%) [11]. Nonetheless, there is important variation in specificity estimates in the literature [11,14,18,19,20,21]. One reason for this is that different studies cover a variety of different PCR protocols to detect different *Leishmania* strains in different geographical regions. However, a recent study from India using the same qPCR protocol as used in this study reported a specificity estimate of 94% (173/184 seropositive non-VL cases from VL endemic regions in India tested negative on qPCR) [12]. Another explanation for the wide range of specificity estimates of molecular tools in the existing literature could be the variation in control groups used. Literature shows that a varying proportion of non-VL cases living in VL-endemic areas can be PCR positive without VL-suggestive symptoms, suggesting that a positive PCR test should be interpreted as a marker of infection rather than disease [12,22,23,24,25]. This is supported by studies reporting the excellent specificity of PCR when using non-endemic controls [22]. On the other hand, not all infected individuals are positive with PCR, as illustrated in our study observing positive qPCR results in only one in 36 (2.8%) rK39 RDT-positive non-VL cases. In addition, the parasitemia of the incident VL cases in our study was generally very high, with a median value of 6844 PGE/mL blood. A 2014 Indian study, using the same qPCR methodology as used in this study, suggested that qPCR can differentiate between asymptomatic infection and (symptomatic) disease, proposing 5 PGE/mL as the cut-off to differentiate between both groups [26]. Among the VL cases identified at the PHC level, we found only four individuals categorized as qPCR-positive to be below this threshold; all the others categorized as qPCR-positive (66/70 = 94%) had a parasitemia above this threshold, which further supports their status as true VL cases. A third explanation of the presumed lack of specificity of PCR in some studies could be the use of a sub-optimal reference test leading to true cases being missed and falsely being categorized as non-diseased, resulting in a systematic underestimation of the specificity of other tests [11]. The sensitivity of parasitology on splenic aspirates is not perfect. While it is generally accepted that the sensitivity is above 90% [27,28], some authors have suggested that the real sensitivity might be lower, and have emphasized the need to adjust for this with complementary analyses when validating other tests [6,29].

Concerns with regard to the decreasing validity of the diagnostic algorithm as a result of declining VL incidence have been previously raised by other authors in calls for more accurate diagnostic tests on the path to the elimination of the disease [2,3]. Although our study suggests that there are only 5% of (potentially) false positives when using the diagnostic algorithm at present, it will be important to continue monitoring this in the years to come in order to avoid misdiagnosis and mistreatment of the patient, as well as wasting of limited resources during reactive measures such as index-case based active case finding or vector control activities in response to each new VL diagnosis. Molecular confirmation of newly diagnosed VL cases in sentinel sites could provide a non-invasive and feasible approach to this aim.

Nonetheless, other concerns with regard to the current diagnostic algorithm remain unchanged. The fact that a patient can only meet the clinical criteria for testing after more than two weeks of fever, by definition, causes a minimum two-week diagnostic delay. Modelling studies suggest that a reduction of this timeframe could have a marked impact on incidence [30]. The inclusion of splenomegaly in the case definition of a suspected VL case further delays the diagnosis, as this often only develops slowly, over a period of weeks to months [31]. In addition, as serological tests do not differentiate between (asymptomatic) infection and disease, they can only be used in strict combination with a clinical algorithm, which complicates their use. Similarly, they don’t differentiate between current or past disease, and can remain positive for years after a VL episode, making them unsuited for the diagnosis of relapse or as tests of cure. Serological tests are also known to be less sensitive for the diagnosis of VL in immunocompromised individuals, which can hinder the detection of VL in these patients in routine settings [32]. The search for more specific diagnostics that can be used as stand-alone tests to detect VL in all patients remains relevant.

## Figures and Tables

**Figure 1 diagnostics-12-00670-f001:**
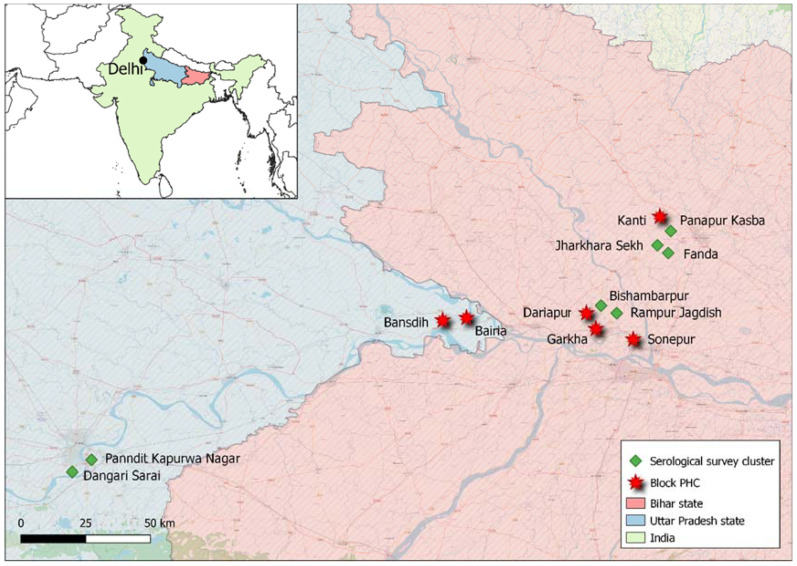
Geographical location of study sites. PHC, primary health care center.

**Table 1 diagnostics-12-00670-t001:** Participant characteristics and qPCR results of incident VL patients. UP, Uttar Pradesh. NA = not applicable.

		Participant Characteristics (*n* = 74)		Results	
State	Block	Median Age (y) (IQR)	Males (*n* (%))	PCR Available (*n*)	PCR Positive (*n* (%))
Bihar	Dariapur	32 (16–50)	29 (57%)	51	49 (96%)
Bihar	Garkha	NA	NA	0	NA
Bihar	Sonepur	24 (16–40)	9 (75%)	12	10 (83%)
Bihar	Kanti	30 (30–30)	1 (100%)	1	1 (100%)
UP	Bairia	23 (15–32)	5 (50%)	10	10 (100%)
UP	Bansdih	NA	NA	0	NA
Overall		29 (16–41)	44 (61%)	74	70 (95%)

**Table 2 diagnostics-12-00670-t002:** Characteristics and results of apparently healthy individuals (non-VL cases). IQR, interquartile range; UP, Uttar Pradesh; Panndit K.N., Panndit Kapurwa Nagar; CE, currently endemic cluster; PE, previously endemic cluster; NE, non-endemic cluster.

			Participant Characteristics (*n* = 15,422)		Results	
	State	Cluster	Median Age (y) (IQR)	Males (*n* (%))	rK39 RDT Available (*n*)	rK39 RDT Positive (*n* (%))
CE	Bihar	Rampur Jagdish	18 (10–38)	1220 (44%)	2772	6 (0.22%)
	Bihar	Bishambarpur	24 (12–24)	1281 (46%)	2756	17 (0.62%)
PE	Bihar	Jhakara Sekh	20 (10–39)	1152 (48%)	2417	2 (0.08%)
	Bihar	Fanda	22 (11–40)	1384 (47%)	2944	12 (0.41%)
	UP	Panndit K.N.	22 (12–37)	311 (46%)	679	2 (0.29%)
NE	Bihar	Panapur Kasba	28 (14–48)	385 (49%)	787	0 (0.00%)
	UP	Dangari Sarai	25 (15–42)	1485 (48%)	3067	0 (0.00%)
		Overall	23 (12–40)	7219 (47%)	15,422	39 (0.25%)

## Data Availability

The data supporting the findings of this study are retained at the Department of Medicine, Institute of Medical Sciences, Banaras Hindu University (BHU), and will not be made openly accessible due to ethical and privacy concerns. Data can, however, be made available to the scientific community immediately following the publication of this manuscript, and after approval of a motivated and written request to BHU at tmrcbhu@gmail.com.

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
