# Peer review of "Diagnosis of Visceral Leishmaniasis in an Elimination Setting: A Validation Study of the Diagnostic Algorithm in India"

_diagnostics, 2022, doi:10.3390/diagnostics12030670_

Round 1
Reviewer 1 Report
The manuscript describes the validation of the positive predictive value (PPV) of the rK39 rapid diagnostic test (RDT) for visceral leishmaniasis in current low-endemic setting in India. This is a much needed validation to warrant a solid interpretation and conclusion of the Kala-azar elimination initiative program that was implemented on the Indian subcontinent in 2005.
The manuscript is well written and the authors presented a compelling amount of data to support the conclusion that a combination of suggestive clinical symptoms with a positive rK39 RDT is a valid algorithm to be used in current low endemic settings in India. Thus, an important tool for the Kala-azar elimination initiative program.
No modification of the manuscript is necessary.
Author Response
We would like to thank the reviewer for this positive appraisal.
Reviewer 2 Report
General comments:
The manuscript addresses an important aspect of routine diagnosis used in VL elimination setting in South Asia.
rK39 RDT remains as an essential tool in VL endemic settings in the region and its sensitivity and specificity are critically important for its efficient and effective utility towards achieving the goals of disease elimination in the region.
While this manuscript has tried to address the above important aspects, it seems to have fallen short in its design to derive valid results towards determining the specificity indicators or negative predictive value. The justification for the study is based on the premise that when the pre-test probability of VL is low (in an elimination setting), a highly sensitive and specific diagnostic tool is required, hence the need to test (and re-validate) the performance of rK39 in such a setting.
However, none of the rK39 negatives have been tested against the gold standard used in the study (qPCR). Furthermore, separating out and describing an area with low case burden, among the study sites hasn’t added anything much to the findings since the numbers tested from that location are very low (n=11) to derive any specific conclusions or make any comments.
Specific comments:
- The term ‘healthy’ used in the abstract and rest of the manuscript (where relevant) should be replaced by ‘apparently healthy’ individuals
- Figure 1 : should be improved by inclusion of standard elements of a figure to make it more meaningful
- Data analysis: calculation of positive predictive value and any other measures that are/ need to be included e.g. negative predictive, sensitivity/specificity values should be elaborated.
Minor comment:
Overall, the manuscript is fairly well-written. However, a few grammatical errors were noted, which would need to be corrected.
Author Response
Please see the attachment.
Author Response:
REPLY TO REVIEWERS
Manuscript reference number: diagnostics-1596531
Title: rK39 Rapid Diagnostic Test for diagnosis of visceral leishmaniasis in an elimination setting: a validation study in India
Dear Reviewer,
I am pleased to submit a revised version of our original research article entitled “rK39 Rapid Diagnostic Test for diagnosis of visceral leishmaniasis in an elimination setting: a validation study in India” (manuscript reference number: diagnostics-1596531).
We have responded to the comments made by the Reviewer point by point and have adapted the manuscript where needed. We hope to have addressed all concerns of the Reviewer to his/her satisfaction.
Thank you for your consideration.
Sincerely,
Kristien Cloots
Unit of Mycobacteria and Neglected Tropical Diseases
Department of Public Health
Institute of Tropical Medicine, Antwerp, Belgium
Response to Reviewer 2 Comments
Point 1: The manuscript addresses an important aspect of routine diagnosis used in VL elimination setting in South Asia. rK39 RDT remains as an essential tool in VL endemic settings in the region and its sensitivity and specificity are critically important for its efficient and effective utility towards achieving the goals of disease elimination in the region. While this manuscript has tried to address the above important aspects, it seems to have fallen short in its design to derive valid results towards determining the specificity indicators or negative predictive value. The justification for the study is based on the premise that when the pre-test probability of VL is low (in an elimination setting), a highly sensitive and specific diagnostic tool is required, hence the need to test (and re-validate) the performance of rK39 in such a setting. However, none of the rK39 negatives have been tested against the gold standard used in the study (qPCR).
Response 1: We would like to emphasize that the primary objective of this study was to assess the performance of the diagnostic algorithm as a whole – i.e. clinical symptoms in combination with a positive rK39 RDT, not of the rK39 RDT as such. Our aim was to provide an answer to the growing concerns of high proportions of false positive VL diagnoses, rather than to evaluate the accuracy of the rK39 RDT as a standalone tool for diagnosis of VL. We have therefore focused on assessing the positive predictive value of the diagnostic algorithm in the current setting, to obtain information on the proportion of individuals who might have wrongly been provided the diagnosis of VL, based on their suggestive clinical picture and a positive rK39 RDT. The negative predictive value (NPV) of the diagnostic algorithm was not assessed in this study, as, contrary to the PPV, the NPV can be expected to be extremely high in the current low prevalence setting, and would tend to further improve with declining case loads. Therefore, there are no reasons to assume that the NPV might be problematic at present.
Although not the primary objective of this study, we included specificity estimates of the rK39 RDT to support and (partly) explain why the PPV of the diagnostic algorithm is higher than might be expected in light of the low VL prevalence. Indeed, no qPCR testing was performed on rK39 RDT negative individuals. However, as none of the more than 15,000 rK39 RDT negative participants in the serosurvey reported any VL-suggestive clinical symptoms, they would – by definition - not have fulfilled the requirements for a VL case, even in case of a positive qPCR test. We therefore believe that they can all be rightfully classified as non-VL cases, providing information on the specificity of the rK39 RDT.
We have reformulated the abstract, data analysis, results, discussion part, as well as the title of the manuscript to better reflect our primary focus on the PPV of the diagnostic algorithm as a whole and not of the rK39 RDT as such. We no longer mention the PPV of the rK39 RDT among healthy individuals, but instead formulate only the specificity estimates of the test. Following the same reasoning, we have adapted Table 2 to present the characteristics of all the participants of the serological survey, not just of the rK39 RDT positive individuals, in order to focus on the specificity estimates rather than on the PPV of the rK39 RDT. In addition, we have added a sentence specifically stating why the NPV of the diagnostic algorithm was not part of the study (lines 67-68): ‘On the other hand, the opposite is true for the negative predictive value, which tends to improve with declining disease prevalence.’ This way, we hope to have sufficiently addressed the reviewer’s comment.
Point 2: Furthermore, separating out and describing an area with low case burden, among the study sites hasn’t added anything much to the findings since the numbers tested from that location are very low (n=11) to derive any specific conclusions or make any comments.
Response 2: We agree that the distinction between the different study sites does not add much to the interpretation of the findings, due to the low numbers. We have therefore adapted the manuscript to this end, and no longer make this distinction.
Point 3: The term ‘healthy’ used in the abstract and rest of the manuscript (where relevant) should be replaced by ‘apparently healthy’ individuals.
Response 3: This was adapted throughout the manuscript.
Point 4: Figure 1 should be improved by inclusion of standard elements of a figure to make it more meaningful.
Response 4: Figure 1 has been adapted to include more geographical reference points, such as rivers and roads. If additional information should be included on the map, we kindly request the reviewer to formulate more specific suggestions for us to incorporate?
Point 5: Data analysis: calculation of positive predictive value and any other measures that are/ need to be included e.g. negative predictive, sensitivity/specificity values should be elaborated.
Response 5: Calculation of the specificity of the rK39 RDT was added on lines 157-159: ‘Specificity of the rK39 RDT was calculated as the proportion of apparently healthy individuals living in currently endemic, previously endemic, or non-endemic areas who were negative by rK39 RDT.’ In addition, we have reformulated the calculations of the PPV to better clarify the focus of the article on the diagnostic algorithm rather than on the rK39 RDT as such (lines 153-157): ‘The positive predictive value (PPV) of the diagnostic algorithm was calculated as the proportion of qPCR positive individuals among incident VL cases diagnosed at PHC level according to the diagnostic algorithm of suggestive clinical symptoms in combination with a positive rK39 RDT.’ As explained in a previous response, we have not included NPV calculations for the diagnostic algorithm, as there is no increasing reason for concern in light of the low disease prevalence. As stated previously, our article focuses on the diagnostic algorithm as a whole rather than on the rK39 RDT as such; therefore, no sensitivity estimates of the rK39 RDT were included.
Reviewer 3 Report
I carefully checked the manuscript titled rK39 Rapid Diagnostic Test for diagnosis of visceral leishmaniasis in an elimination setting: a validation study in India.
The authors described the PPV and PPN of rK39 RDT in endemic areas of India using as reference standard the qPCR for diagnosed and suspected patients.
As they also declare in their manuscript (page 6 lines 218-224) the use of a qPCR as reference standard is not the best choice. Moreover, following different PCR protocols the accuracy of this test can change, authors should also state this and indicate the proper references.
Indeed, what authors found above the Findings among asymptomatic rK39 RDT positive individuals cannot be presented as sure data. It can only be presented as descriptive data because a more accurate study has to be done.
It is very hard to say that PPV and PPN for healthy and asymptomatic patients was xx%, it is well known that antibodies can persist in the blood for a long time as well as PCR in peripheral blood is not that much sensitive.
Despite authors are using (potentially) they cannot affirm that their study suggests that there are only 5% of false positives when using the diagnostic algorithm. Authors should change the discussion and present the data as descriptive; unfortunately, no more hypotheses can be done with the results that they are presenting.
Authors are referring almost to articles from their group(s), they should improve the literature by checking other available studies and refereeing to them. As an example, but not limited to:
- DOI 10.1016/S2222-1808(13)60003-1
- DOI 10.1371/journal.pone.0185606
- DOI 10.1590/S0036-46652013000200006
- DOI 10.1128/JCM.00132-21
Author Response
Please see the attachment.
Author Response:
REPLY TO REVIEWERS
Manuscript reference number: diagnostics-1596531
Title: rK39 Rapid Diagnostic Test for diagnosis of visceral leishmaniasis in an elimination setting: a validation study in India
Dear Reviewer,
I am pleased to submit a revised version of our original research article entitled “rK39 Rapid Diagnostic Test for diagnosis of visceral leishmaniasis in an elimination setting: a validation study in India” (manuscript reference number: diagnostics-1596531).
We have responded to the comments made by the Reviewer point by point and have adapted the manuscript where needed. We hope to have addressed all concerns of the Reviewer to his/her satisfaction.
Thank you for your consideration.
Sincerely,
Kristien Cloots
Unit of Mycobacteria and Neglected Tropical Diseases
Department of Public Health
Institute of Tropical Medicine, Antwerp, Belgium
Response to Reviewer 3 Comments
Point 1: I carefully checked the manuscript titled rK39 Rapid Diagnostic Test for diagnosis of visceral leishmaniasis in an elimination setting: a validation study in India. The authors described the PPV and PPN of rK39 RDT in endemic areas of India using as reference standard the qPCR for diagnosed and suspected patients. As they also declare in their manuscript (page 6 lines 218-224) the use of a qPCR as reference standard is not the best choice. Moreover, following different PCR protocols the accuracy of this test can change, authors should also state this and indicate the proper references.
Response 1: We fully agree that the accuracy of qPCR testing depends on the exact method followed, and that this was not sufficiently mentioned previously. We have therefore now added a few sentences specifically emphasising this (lines 249 – 252): ‘Nonetheless, there is important variation in specificity estimates in the literature […]. One reason for this is that different studies cover a variety of different PCR protocols to detect different leishmania strains in different geographical regions.’ In addition to the more general statements about molecular tools overall, we now mention the specific sensitivity and specificity estimates available from studies using the exact qPCR procedures followed in this study as well (performed by the same laboratory), which were excellent.
- Lines 214 - 217: ‘Nonetheless, a more recent study conducted in Bangladesh, as well as two recent studies in India using the same qPCR protocol as used in the current study (conducted by the same laboratory), reported a 100% sensitivity of qPCR for the diagnosis of VL […].’
- Lines 252 – 256: ‘However, a recent study from India using the same qPCR protocol as used in this study reported a specificity estimate of 94% (173/184 apparently healthy seropositive individuals from VL endemic regions in India tested negative on qPCR) […]. Since these asymptomatic individuals all showed serological signs of infection with leishmania, the specificity estimate is likely to be an underestimation.’
Point 2: Indeed, what authors found above the Findings among asymptomatic rK39 RDT positive individuals cannot be presented as sure data. It can only be presented as descriptive data because a more accurate study has to be done. It is very hard to say that PPV and PPN for healthy and asymptomatic patients was xx%, it is well known that antibodies can persist in the blood for a long time as well as PCR in peripheral blood is not that much sensitive.
Response 2: We fully acknowledge that antibodies can persist in the blood for a long time. This is exactly the reason why as a main objective we assessed the PPV of the diagnostic algorithm (positive RDT + clinical symptoms), and not the PPV of the rK39 RDT as such. A positive antibody test is merely a sign of infection, which is meaningless for diagnosis of VL in absence of clinical signs of disease. This is supported by our findings of high qPCR positivity rates among symptomatic RDT positives (95%) compared to asymptomatic RDT positives (2.8%).
With regard to the concerns raised about the sensitivity of the reference test used, we would like to emphasize that, although accuracy of molecular methods can indeed vary depending on the exact protocol used, qPCR on peripheral blood targeting leishmania kDNA is known to be highly sensitive. Two recent studies conducted in India using the exact same qPCR procedure as was used in this study, reported a 100% sensitivity of the qPCR [1, 2]. Moreover, a 2021 review [3] concluded the following: ‘To enable Leishmania detection, a molecular target should have high abundance and this criteria is best achieved by (i) kinetoplast mini-circle DNA (kDNA), present as 1,000s copies per cell in all Leishmania sp (Salotra et al., 2001; Mary et al., 2004; Mary et al., 2006; Verma et al., 2010) and (ii) 18s rRNA (Deborggraeve et al., 2008; Mehrotra et al., 2011; Srivastava et al., 2011). As a diagnostic tool for VL, the kDNA based qPCR or real-time PCR (Table 2) has stood the test of time (Mary et al., 2006; Verma et al., 2010; Sudarshan et al., 2011; Abbasi et al., 2013; Sudarshan et al., 2015; Hossain et al., 2017).’ We are therefore confident that the the qPCR method used as a reference test in this study was an appropriate and sufficiently sensitive choice.
However, we do acknowledge that the original formulation of the Findings among asymptomatic rK39 RDT positive individuals was suboptimal. As per the reviewer’s suggestions, we have reformulated this paragraph and no longer mention the PPV of the rK39 RDT among asymptomatic rK39 RDT positive individuals. Instead, we redirected the focus of this paragraph towards the specificity estimates of the rK39 RDT that can be drawn from the serological survey. In line with this change, we have adapted Table 2, which no longer focuses on the rK39 RDT positive individuals, but now instead presents the characteristics and findings among the healthy (non-VL diseased) participants. Hereby, we hope to have sufficiently addressed the reviewer’s concerns.
Point 3: Despite authors are using (potentially) they cannot affirm that their study suggests that there are only 5% of false positives when using the diagnostic algorithm. Authors should change the discussion and present the data as descriptive; unfortunately, no more hypotheses can be done with the results that they are presenting.
Response 3: Although we acknowledge that the qPCR test on peripheral blood is not traditionally used as the gold standard for confirmation of VL, we would like to clarify our choice for this reference test. As clarified in the response to the previous comment, qPCR on peripheral blood, especially when using kDNA primers, is unambiguously accepted as highly sensitive for the diagnosis of VL throughout the literature, with two recent Indian studies using the same methodology obtaining sensitivities of 100% [1-6]. In general, it’s especially the specificity of molecular methods which is more commonly questioned, with as a main concern that positive results reflect infection rather than disease. However, our study illustrates that not all infected individuals were qPCR positive; only 1 in 36 (2.8%) of rK39 RDT positive individuals tested positive for qPCR. Another recent study using the same qPCR protocol found only 11 out of 184 apparently healthy but infected individuals (either high antibody titers in rK39 ELISA or Direct Agglutination test, or positive on both) to be qPCR positive, reflecting a specificity of 94% [1]. To support our claims for the high specificity of the qPCR method used in this study, we have included additional data on the quantification of the parasitemia in the included VL patients (results section lines 170 – 172, discussion section lines 259 – 266). These results show generally very high parasitemias, with a median value of >6 000 PGE/ml blood. A 2014 Indian study using the same qPCR methodology as used in this study, suggested that qPCR can differentiate between asymptomatic infection and (symptomatic) disease, proposing 5 PGE/ml as the cut-off to differentiate between both groups [7]. In the VL cases identified at PHC level, we found only four individuals categorized as qPCR positive to be below this threshold; all the others (66/70 = 94%) categorized as qPCR positive had a parasitemia above this threshold, which further supports their status as true VL cases. This illustrates that qPCR does not merely reflect infection, but is associated with clinical disease. We therefore believe that the qPCR method used in this study was both an acceptably sensitive and specific reference method to support our conclusion that there seem to be (only) 5% false positives with the diagnostic algorithm in the current setting, and we hope the Reviewer can agree.
Point 4: Authors are referring almost to articles from their group(s), they should improve the literature by checking other available studies and refereeing to them. As an example, but not limited to:
- DOI 10.1016/S2222-1808(13)60003-1
- DOI 10.1371/journal.pone.0185606
- DOI 10.1590/S0036-46652013000200006
- DOI 10.1128/JCM.00132-21
Response 4: Our focus on the Indian subcontinent indeed guided us towards several studies of our own research group. We have now adapted the manuscript to include a wider range of available studies from other research groups and other global regions as well. Studies not conducted by our research group which were added to the reference list are:
- Gass et al. 2020 [8]
- Hossain et al. 2017 [9]
- Salotra et al. 2001 [4]
- Da Costa Lima et al. 2013 [10]
- Silva et al. 2013 [11]
- Abbasi et al. 2013 [12]
- Moulik et al. 2021 [3]
References:
- Singh, O.P., et al., Xenodiagnosis to evaluate the infectiousness of humans to sandflies in an area endemic for visceral leishmaniasis in Bihar, India: a transmission-dynamics study. Lancet Microbe, 2021. 2(1): p. e23-e31.
- Sudarshan, M., et al., A Correlative Study of Splenic Parasite Score and Peripheral Blood Parasite Load Estimation by Quantitative PCR in Visceral Leishmaniasis. J Clin Microbiol, 2015. 53(12): p. 3905-7.
- Moulik, S., S. Sengupta, and M. Chatterjee, Molecular Tracking of the Leishmania Parasite. Front Cell Infect Microbiol, 2021. 11: p. 623437.
- Salotra, P., et al., Development of a species-specific PCR assay for detection of Leishmania donovani in clinical samples from patients with kala-azar and post-kala-azar dermal leishmaniasis. J Clin Microbiol, 2001. 39(3): p. 849-54.
- Mary, C., et al., Quantification of Leishmania infantum DNA by a real-time PCR assay with high sensitivity. J Clin Microbiol, 2004. 42(11): p. 5249-55.
- Verma, S., et al., Quantification of parasite load in clinical samples of leishmaniasis patients: IL-10 level correlates with parasite load in visceral leishmaniasis. PLoS One, 2010. 5(4): p. e10107.
- Sudarshan, M. and S. Sundar, Parasite load estimation by qPCR differentiates between asymptomatic and symptomatic infection in Indian visceral leishmaniasis. Diagn Microbiol Infect Dis, 2014. 80(1): p. 40-2.
- Gass, K., Time for a diagnostic sea-change: Rethinking neglected tropical disease diagnostics to achieve elimination. PLoS Negl Trop Dis, 2020. 14(12): p. e0008933.
- Hossain, F., et al., Real-time PCR in detection and quantitation of Leishmania donovani for the diagnosis of Visceral Leishmaniasis patients and the monitoring of their response to treatment. PLOS ONE, 2017. 12(9): p. e0185606.
- da Costa Lima, M.S.J., et al., Sensitivity of PCR and real-time PCR for the diagnosis of human visceral leishmaniasis using peripheral blood. Asian Pacific Journal of Tropical Disease, 2013. 3(1): p. 10-15.
- Silva, L.A., et al., USE OF THE POLYMERASE CHAIN REACTION FOR THE DIAGNOSIS OF ASYMPTOMATIC Leishmania INFECTION IN A VISCERAL LEISHMANIASIS-ENDEMIC AREA. Revista do Instituto de Medicina Tropical de São Paulo, 2013. 55(2): p. 101-104.
- Abbasi, I., et al., Evaluation of PCR procedures for detecting and quantifying Leishmania donovani DNA in large numbers of dried human blood samples from a visceral leishmaniasis focus in Northern Ethiopia. BMC Infect Dis, 2013. 13: p. 153.
Round 2
Reviewer 3 Report
I checked the revised manuscript and appreciate the improvement authors did.
The authors properly addressed all the points and the new title reflects the study. Indeed, the manuscript is good enough for its publication.
A few minor changes are still needed:
- Authors should update the reference of the qPCR methods. The protocol for qPCR cannot be found in the reference they are provided. Following their reference, the protocol can be found at 10.1371/journal.pntd.0003366.
- “apparently healthy individuals” should be called “non-VL cases” I would not use “healthy” even if preceded by "apparently"
-pag 6, lines 195 and 197-199: something went wrong with pdf creation. Please check it.
Author Response
Point 1: Authors should update the reference of the qPCR methods. The protocol for qPCR cannot be found in the reference they are provided. Following their reference, the protocol can be found at 10.1371/journal.pntd.0003366.
Response: We have updated the reference for the qPCR method, now referring to the original reference as suggested by the reviewer.
Point 2: “Apparently healthy individuals” should be called “non-VL cases” I would not use “healthy” even if preceded by "apparently".
Response: This has been adapted throughout the manuscript.
Point 3: pag 6, lines 195 and 197-199: something went wrong with pdf creation. Please check it.
Response: We thank the reviewer for pointing out this error. This was corrected in the current pdf.